# Pharmacy Implementation of a New Law Allowing Year-Long Hormonal Contraception Supplies

**DOI:** 10.3390/pharmacy8030165

**Published:** 2020-09-06

**Authors:** Gelareh Nikpour, Antoinette Allen, Sally Rafie, Myung Sim, Radhika Rible, Angela Chen

**Affiliations:** 1Department of OBGYN, Southern California Permanente Medical Group, Downey, CA 90242, USA; 2Department of OBGYN, School of Medicine, University of Pennsylvania, Philadelphia, PA 19104, USA; Antoinette.Allen@pennmedicine.upenn.edu; 3Department of Pharmacy, UC San Diego Health, San Diego, CA 92093, USA; srafie@health.ucsd.edu; 4Med-GIM & HSR, UC Los Angeles, Los Angeles, CA 90095, USA; msim@mednet.ucla.edu; 5Department of OBGYN, UC Los Angeles, Los Angeles, CA 90095, USA; rrible@mednet.ucla.edu (R.R.); angelachen@mednet.ucla.edu (A.C.)

**Keywords:** year-long, hormonal contraceptive method, birth control

## Abstract

Background: Prescription hormonal contraceptive methods are vital to prevention of unplanned pregnancies. New legislation among 23 states has expanded access to contraception. In California, a 2017 law requires pharmacists to dispense year-long supplies of contraception and insurance plans to cover it upon patients’ request. This study assesses pharmacist knowledge of this new law 6 months after enactment. Methods: From July to November 2017, a random selection of 600 community pharmacies were called requesting a pharmacist (n = 532, 88.7% response). Pharmacists were asked if they had heard of the new law, if they would dispense a year-long supply to cash-pay, privately or publicly insured patients, and what they perceived as obstacles to dispensing year-long supplies. Results: Awareness of this law was assessed through these surveys. Most pharmacists responded they would dispense year-long supplies to cash-pay patients, regardless of knowledge of the new law (81% of “knew”, 70% of “did not know”, *p* = 0.1046). The top two perceived obstacles were insurance reimbursement (55.8%) and store policy (13.4%). Conclusion: Despite a new law requiring insurance coverage of a year-long supply of prescription birth control, most pharmacists were unaware at six months after the policy went into effect. Of those who were aware, the majority did not clearly understand it. Compliance among insurance plans is unknown. There was no implementation plan or awareness campaign for the new law.

## 1. Introduction

Access to modern methods of contraception is essential in the prevention of unplanned and unintended pregnancies. Hormonal contraceptive methods are commonly used by reproductive-aged women to prevent unintended pregnancies; however, access and affordability remain obstacles, particularly in under-insured and under-resourced communities. Notably the rate of unintended pregnancies in women who live below the United States national poverty line is five times the rate in women 200% above the poverty line [1]. The Affordable Care Act was passed in an attempt to remove obstacles to access by decreasing or eliminating patient co-pays for prescription contraception services and supplies for many insured patients. Despite progress, the ability to obtain these products from pharmacies remains a prominent issue as nearly 20 million women in the United States live in contraceptive deserts, lacking access to all forms of birth control [2]. Having to travel far distances every one to three months thus becomes a significant barrier to accessing hormonal contraceptive methods.

While long-acting reversible contraceptives, such as the intrauterine device and the subdermal implant (accounting for 8% and 6% of contraception use, respectively) are the most effective methods of birth control and reduce the number of times women need to be seen by a clinician, a higher percentage of women use short-term methods such as oral contraceptive pills (27%), the contraceptive patch (1%), and the vaginal ring (2%) [3]. These women are, therefore, dependent on their physicians, pharmacists and insurance companies for regular access to their chosen form of hormonal birth control. Insurance plans have historically covered one- or three-month supplies of birth control to be dispensed from community pharmacies, creating a potential obstacle to obtaining refills supplies and continuing birth control use.

Some patients face additional challenges in obtaining their supplies. In a study performed by Barber et al. in 2019, while African American women tend to live closer to pharmacies than their white counterparts, those pharmacies are more likely to be community-based, independent pharmacies that have limited opening hours, less visible contraceptive information, more difficult access to condoms (behind the counter) and fewer self-checkout options [2]. This adds nuance to the conventional definition of “contraceptive desert” and lays the groundwork for a need for statewide policies that are effectively implemented to remove obstacles to obtaining birth control, as there may be more barriers impeding access to contraception than the presence of a local pharmacy.

A study evaluating the effects of year-long birth control supplies as dispensed by clinics affiliated with California’s Family Planning, Access, Care and Treatment (PACT) program—a state program which provides comprehensive family planning services ranging from providing birth control methods (long-acting reversible methods, birth control pills/patch/ring, sterilization), abortion care, testing and treatment for sexually transmitted disease, and cervical cancer screening, for low-income residents—found a 30% odds reduction in unplanned pregnancy as compared with dispensing just one or three packs [4,5]. In 2016, the Centers for Disease Control and Prevention recommended removing barriers such as “restrictions on number of pill packs dispensed at one time” to allow easier access to prescription hormonal contraceptive methods [6]. Seeking to make this evidence-based practice widely available in the state, California Senate Bill 999 (CSB 999) was passed and signed into law in 2016, ensuring that women can receive up to a year-long supply of their chosen prescription contraceptive method, allowing pharmacists to dispense and mandating insurance companies to cover these prescriptions. This new law was enacted on 1 January 2017 [7]. Similar legislation has been passed in 21 other states including Washington D.C. and in most cases, the legislation allows for year-long supplies [8]. All forms of self-administered prescription birth control methods—including the combined and progestin-only oral contraceptive pills, the vaginal ring, and the hormonal patch—were included in the law providing a year-long supply.

With the initiation of this new law, women will be able to access their selected contraception more readily, decreasing the rate of method discontinuation and subsequently reducing the number of unplanned, unwanted pregnancies. The rate-limiting steps will depend on when and how pharmacies and insurance companies develop protocols and personnel training to ensure year-long supplies are being dispensed. Furthermore, patients and prescribers will need to be informed of these new measures so they are aware of extended access.

Although the policy came into effect on 1 January 2017, it is unclear whether pharmacies are filling year-long supplies of birth control upon patient request. For community pharmacies to provide greater access to hormonal contraception, pharmacists must have knowledge of this new policy. This study evaluates how pharmacies in California respond to anticipating patient requests for year-long supplies of their prescribed contraception method. This study also seeks to identify pharmacist perceived barriers to dispensing year-long supplies of hormonal contraception.

## 2. Materials and Methods

This project is a cross-sectional survey of pharmacies in California using a “secret shopper” technique. “Secret shopper” is an observational research methodology used to evaluate internal consistency among entities to identify areas for process improvement by using an individual surveyor with acting as a care prescriber asking about access to year-long self-administered prescription birth control for her patients. This study was reviewed by the institutional review board of the School of Medicine at the University of California, Los Angeles, and deemed to be of minimal risk with the assumption that no practical, non-deceptive alternative to obtaining unbiased results was appropriate for the study design. The “secret shopper” methodology offers simulation under controlled conditions of a live encounter with a medical practice, thus permitting unbiased assessment of the payment method variables which are thought to be relevant to access to contraception. Role theory informed the study design with the a priori hypothesis that topics discussed and comfort with dispensing hormonal birth control would vary based on knowledge of CSB 999. Role theory is a sociological concept that hypothesizes that all interactions are played out in a circumscribed set of categories where each role has widely recognized rules, norms and expectations with which each participant must act out and interact. Role theory is critical in qualitative research, especially with secret/mystery shopper methodology as the whole basis of the investigation rests on the subject performing their predefined role and the investigator challenging that role while remaining within the bounds of what is considered “socially acceptable” for particular interaction (e.g., moviegoer and usher telling you the weather forecast) [9]

Interpretation of CSB 999 was derived from the leginfo.legislature.ca.gov website where the complete verbiage of the new law is delineated. CSB 999 is a bill that amends and adds to previous laws: amendment to Section 4064.5 of the Business and Professions Code; amendment to Section 1367.25 of the Health and Safety Code; amendment to Section 10123.196 of the Insurance Code; addition to Section 14000.01 to the Welfare and Institutions Code. With the passage of CSB 999, pharmacists can dispense twelve pill packs per patient request even if the prescription is written for a 3-month supply with refills. The final bullet point of the bill’s summary states “the intent of the Legislature to expand on California’s existing contraceptive coverage policy by requiring all health care service plans and health insurance policies, including both commercial and Medi-Cal managed care plans, to cover a year-long supply [at one time] of a prescribed Food and Drug Administration (FDA)-approved contraceptive, such as the ring, the patch, and oral contraceptives.” Prior to these amendments, section 4064.5 of the Business and Profession Code reads, “[a] pharmacist may dispense not more than a 90-day supply of a dangerous drug other than a controlled substance pursuant to a valid prescription that specifies an initial quantity of less than a 90-day supply followed by periodic refills of that amount.” With one of the amendments introduced in CSB 999, self-administered hormonal contraception is excluded from this category and a further statement explains that “[a] pharmacist shall dispense, at a patient’s request, up to a year-long supply of an FDA-approved, self-administered hormonal contraceptive pursuant to a valid prescription that specifies an initial quantity followed by periodic refills.” Section 10123.196 of the Insurance Code is amended to read goes on to clarify that insurance policies are required to cover the cost of a year-long supply being dispensed at one time. Prior to the passage of CSB 999, pharmacists could only dispense a year-long supply if the physician specifically wrote for 12 pill packs. With the passage of CSB 999, pharmacists can dispense 12 pill packs even if the prescription is written as 1 pack plus 12 refills per patient request. For pharmacy selection, we reviewed 19,998 pharmacies listed in California on the California Board of Pharmacy website in May 2017, excluding all pharmacies with revoked or cancelled licenses, thus removing 13,013 pharmacies from the list. We also excluded pharmacies that were affiliated with managed care or publicly funded networks such as Kaiser Permanente, removing an additional 1101 pharmacies from the list. From 5874 remaining community pharmacies with active licenses, we randomly selected 20 pharmacies for a power calculation.

This pilot study was conducted using the same survey script we used for the primary study. Among randomly selected 20 pharmacists, 100%, 42.9% and 28.6% answered they would dispense a year-long supply to cash-pay, privately and publicly insured patients, respectively. Using these proportions as preliminary data, we powered (80% power) for all pairwise comparisons at the 0.017 significance level. We found that in order to have a power of 80%, 581 pharmacists would need to be surveyed. The 20 pharmacies that were included in this initial pilot study were subsequently excluded from the final sample set bringing the number down to 5854. Of the 5854 pharmacies that met inclusion/exclusion criteria, 600 were randomly selected for inclusion in the current study (Figure 1) and 5254 were not included in the study.

Pharmacies were stratified by income percentiles using their zip codes. Income percentiles were derived from statiticalatlas.com using the data provided by US Census Bureau and then divided into 20th, 40th, 60th, 80th and 95th percentiles (Table 1).

From July 2017 through November 2017, five trained study staff surveyed a pharmacist at each location using a “secret-shopper” telephone survey. Study staff surveyed pharmacists to assess if they were aware and currently complying with the law allowing dispensing of a one-year supply of hormonal contraception to patients upon request. Each call was attempted 3 times and limited questions were used in the script (Table 2, scripted survey questions). After the third missed call, the pharmacy was listed as non-responsive. Upon calling the pharmacy, the survey staff requested to speak directly to a pharmacist. The survey staff introduced themselves as staff at local obstetrics and gynecology (OBGYN) clinics calling with general questions about prescribing birth control. No patient information was disclosed during the survey.

The survey was divided into three groups: one group being asked about patients who were cash-pay; one group being asked about patients who were privately insured; and one group being asked about patients who were publicly insured. Pharmacies were then randomized into these three groups-cash-pay, privately insured, or publicly insured. In all groups, the secret shopper would request to speak to a pharmacist directly and ask if he or she had heard of the California state law allowing pharmacists to dispense an entire year’s supply of prescription birth control upon patient request. “Have you heard of the new law allowing pharmacists to dispense a year-long supply of medications?” Then, pharmacists were each asked a second question depending on the group to which they were randomized. The pharmacist randomized to the cash-pay group was specifically asked “Would you be able to dispense a year-long supply of birth control to cash-pay patients?” The pharmacists randomly selected to the privately insured and publicly insured would be asked whether they would dispense to privately insured or publicly insured patients, respectively. (“Would you be able to dispense a year-long supply of birth control to privately insured patients?” “Would you be able to dispense a year-long supply of birth control to publicly insured patients?”) Finally, all pharmacists were asked what they perceived as obstacles to dispensing a year-long supply of birth control to patients (Table 2). Questions 1 and 2A-C were a binary yes or no answer, given a numeric value, 0 for no, 1 for yes, and assessed quantitatively. Using a multiple comparison test for proportions in a 2xc crosstabulation with logistic regression procedures in SAS^®^, to compare whether or not there was a statistical significant between whether pharmacists would dispense a year-long supply of birth control to cash pay versus privately insured, cash pay versus publicly insure, and privately insured versus publicly insured. The pharmacists gave a qualitative response to Question 3. The study staff then reviewed the answers for Question 3 after the survey. This response was analyzed for content and keywords were tagged to allow thematic analysis of the qualitative data. These tagged keywords were then reviewed by the one data reviewer for standardization. (Table 3, obstacles to dispensing year-long birth control). These coded words were counted to allow comments to be quantitatively assessed.

Baseline and demographic characteristics were summarized by standard descriptive summaries such as means and standard deviations for continuous variables such as median area income and percentages for categorical variables such as discrete number of pharmacies. There was no statistical significance between income percentiles and responses to Q2A-C (*p* = 0.78, 0.38, 0.64, respectively).

Survey data were assessed using chi-squared tests to compare statistical significance between insurance type and whether pharmacists would dispense medications. We also used a multivariable logistic regression model to assess whether pharmacists perceived an obstacle in dispensing medications and whether it differed between payment type (public insurance, private insurance and cash pay).

## 3. Results

The “secret shopper” telephone surveys resulted in a response rate of 88.7% (*n* = 532).

Of the 532 pharmacists asked whether they had heard of CSB 999 allowing dispensing of a year-long supply of prescription birth control, 530 responded. The majority of pharmacists had not heard of the law (353, 66.4%), while roughly one-third had heard of it (177, 33.3%.) Table 4 shows the breakdown of survey results for questions 1 and 2A–C. In summary, reviewing each cohort, about 73% would dispense to cash-pay, 55% to privately insured and 45% to publicly insured. There was a statistical significance between dispensing to cash pay as compared to privately insured as well as cash pay compared to publicly insured. When comparing private and public insurance, there was no statistical significance.

When comparing Question 1 (“Have you heard of the new law allowing pharmacies to dispense year-long supplies of birth control?”) to Question 2A–C (“Would you dispense to cash pay, privately-insured or publicly-insured women?”) the results were only statistically significant for the cash-pay group. Within this cohort, of the pharmacists who stated yes they have heard of the law, 81% would dispense; while of the pharmacists who had not heard of the law, 70% would still dispense to cash pay women (*p* = 0.105). Of the cohort asked about dispensing to privately insured patients, 48% of the pharmacists who knew of the law would dispense a year-long supply, while 58% who did not know of the law would dispense (*p* = 0.213). Of the final cohort asked about publicly insured patients, 47% of the pharmacists who knew of the law would dispense a year-long supply, while 43% who did not know of the law would dispense (*p* = 0.687). (Table 5: Yes or no responses to Question 1 compared to yes or no responses to Questions 2A–C.).

Reviewing the qualitative data retrieved by Question 3 (Figure 1), we found that the majority of pharmacists stated that the biggest perceived obstacle for dispensing a year-long supply of birth control at once was whether insurance companies would provide appropriate reimbursement (297 out of 532, 55.35%). The second most commonly perceived obstacle was whether the pharmacy had a protocol or policy allowing for such a large quantity of birth control (71 out of 532, 13.35%). The third most commonly perceived obstacle was whether or not the pharmacy would have an adequate supply for large quantities of birth control (28 out of 532, 5.26%). See Table 2 for breakdown of 10 obstacles and number of pharmacists who stated that was an obstacle).

When comparing the data presented in Question 3 against the three cohorts from Question 2, there were several areas of correlation with statistical significance. A pharmacist was 6.16 times more likely to dispense a one-year supply of birth control to cash-pay compared to privately insured patients (confidence interval (CI) 95%, 3.45–11.02, *p* < 0.001) and 10.07 times more likely to cash-pay compared to publicly insured patients (CI 95%, 5.52–18.35, *p* < 0.0001) when we controlled for the variable that stated whether or not insurance was an obstacle. Furthermore, they were 1.63 times more likely to dispense a one-year supply of birth control to privately insured patients compared to publicly insured patients (CI 95%, 1.04–2.56, *p* < 0.01) when we controlled for the variable that stated whether or not insurance was an obstacle. When we controlled for the variable that stated whether or not store policy was an obstacle, pharmacists were 2.82 times more likely to dispense a one-year supply of birth control to cash pay compared to privately insured patients (CI 95%, 1.68–4.72, *p* < 0.0001) and 4.39 times more likely to cash-pay compared to publicly insured patients (CI 95%, 2.63–7.33, *p* < 0.0001). Similarly, when we controlled for the variable that stated whether or not patient safety was an obstacle, pharmacists were 2.27 times more likely to dispense a one-year supply of birth control to cash pay compared to privately insured patients (CI 95%, 1.44–3.56, *p* < 0.001) and 3.50 times more likely to cash-pay compared to publicly insured patients (CI 95%, 2.23–5.48, *p* < 0.0001). Lastly, we also incorporated store policy, patient safety and insurance in the model along with the main study variable ‘pay type’ and we found that a pharmacist was 5.02 times more likely to dispense a one-year supply of birth control to cash-pay compared to privately insured patients (CI 95%, 2.62–9.63, *p* < 0.0001) and 8.29 times more likely to cash-pay compared to publicly insured patients (CI 95%, 4.28–16.06, *p* < 0.0001) when we controlled for all three variables (whether or not store policy, patient safety and insurance were obstacles). Furthermore, they were 1.65 times more likely to dispense a one-year supply of birth control to privately insured patients compared to publicly insured patients (CI 95%, 1.03–2.65, *p* < 0.05). (Table 6, Logistic regression model control for store policy, patient safety, and insurance).

## 4. Discussion

This study reviewed implementation just 6 months after a new state law at community pharmacies by surveying pharmacists using a secret shopper phone call. When asked whether the pharmacist had heard of CSB 999, only one-third stated that they were aware of it. Furthermore, despite only one-third of pharmacists being aware of the law, knowledge of the law does not appear to impact pharmacists’ perceived ability to dispense the medication. In fact *fewer* pharmacists stated they would dispense a year-long supply of birth control when *knowing* the law than *not knowing* the law (48% versus 58%). This signals an incomplete understanding of CSB 999 and all its provisions. It is also possible that pharmacists wanted to appear knowledgeable about relevant policies when speaking to a representative from a local gynecologist office.

With multivariable assessment, pharmacists noted that cash-pay patients would be able to request and receive their year-long supply significantly more frequently than both privately and publicly insured patients. Cash-pay patients likely bypass an obstacle that insured patients face: coverage from insurance companies. This assumption is aligned with the results of what pharmacists perceived as the major obstacle in dispensing a year-long supply of birth control. Pharmacists noted nine specific obstacles as primary reasons that pharmacists could not dispense year-long supplies of birth control with reimbursement from insurance as the foremost reason. CSB 999 clearly states that all insurance companies are required to cover the expense to dispense a year-long supply of birth control *at one time* to patients upon request. Despite the clear verbiage of this law, pharmacists are particularly concerned that insurance companies will not reimburse pharmacies for a year-long supply. This reflects the critical role of insurance company practices on the ability of pharmacies to provide patient-centered care.

Furthermore, as the implementation of this new law remained incomplete, many chain and local retail pharmacies had not created store policies to educate their staff. In fact, not having a unifying store policy was second to insurance policy coverage (13.4%) as another common reason for not being able to dispense a year-long supply. Despite concerns over store policies, supplies and even patient safety, cash-paying patients were still more likely to receive their year-long supply over privately and publicly insured patients, reinforcing that once again, reimbursement was the primary perceived obstacle by the pharmacists.

Less frequently, but importantly, 3.0% of pharmacists stated that they could not dispense a year-long supply at one time unless the prescription explicitly made that statement. This brings another factor into account, the medical prescribers. In CSB 999, the responsibility of dispensing the medication in year-long supplies was given to the pharmacist, the responsibility for compensation was given to the insurance provider. Even when the prescription is interpreted by the pharmacist as a 90-day supply with refills for 1 year, they are required to provide a full year-long supply at one time if the patient requests it as such. Thus, while the responsibility for dispensing and coverage rests with the pharmacies and insurance companies, prescribers can facilitate these practices by writing prescriptions for a year-long supply and informing patients about the option of requesting a year-long supply of birth control at once.

This study reviewed implementation of the laws amended and created by CSB 999 in a five-month period starting approximately six months after enactment of the bill. One of the drawbacks to this study includes the time-lapse between enactment of the laws and actual implementation. There is a possibility that administrations of chain pharmacies were working on store policies and pharmacist training programs to enact the laws put forth by CSB 999 which have since been implemented. Furthermore, questioning individual pharmacists as representatives for entire chain pharmacies could perhaps miss the efforts made at higher administrative levels of the pharmacies.

Another drawback to this study was that the “secret shopper” methodology is rather inflexible, not allowing significant variation in questions that are “off-script.” Furthermore, no patient information was used and all pharmacists were asked only “hypothetically” if they would be able to dispense year-long supplies at once to patients. Thus, the pharmacist was not able to then run an actual prescription in their system to assess whether or not there would truly be an obstacle. The obstacles listed are thus only perceived obstacles from the pharmacists’ perspective based on the obstacles they experienced in the past with dispensing birth control.

The last drawback to the “secret shopper” model is the possibility of divergence from the script by the surveyor. In several surveys, pharmacists did not respond to questions. Furthermore, the qualitative nature of the answers to Question 3 requires interpretation by the surveyor. There were 25 pharmacists (4.8%) who gave answers that were not applicable to any category and there were 15 pharmacists (2.8%) who were unsure about the obstacles to dispensing a year-long supply of birth control.

Enactment of a law without plans for direct implementation renders the law essentially obsolete. Implementation of new laws requires publicity and timelines. While the best-established contraception data focus on long-acting reversible contraception (LARC) with low failure rates, important comparisons can be made with a longer supply of prescription birth control methods such as the pill, patch, and ring. By identifying obstacles and implications of CSB 999, our study adds to the data supporting the importance of state supported family planning care and decreases in unplanned pregnancies and associated health care spending.

As a next step, future research should evaluate interval assessment of implementation of CSB 999 since 2017 in California and survey the other states that have enacted similar policies. Chain pharmacies should be reviewed at an administrative level to assess plans for implementation and education of pharmacy staff. Additionally, a review of insurance companies should investigate the internal obstacles they face in abiding by this new California state law.

## 5. Conclusions

Given that providing access to contraception in year-long supplies has been shown to decrease the rate of unintended pregnancies, this study of an important law in 2017 has many implications. In today’s polarized political climate, laws ensuring that patients can have access to contraception is vital to the healthcare of reproductive-aged women. Furthermore, as social distancing and sheltering at home during the COVID-19 pandemic continues to sweep the country, patients and medical establishments are searching for ways to limit clinical exposures to infectious disease. Providing year-long supplies is an excellent means to this end, potentially reducing unnecessary pharmacy visits monthly or every 3 months and also significantly preventing unplanned and undesired pregnancies.

As access to management of unplanned and unwanted pregnancies, either through abortion or prenatal care, is limited by state and federal legislation, prevention of these pregnancies through contraception becomes more important than ever.

This study reinforces the importance of implementation of the extended contraceptive supply laws.

## Figures and Tables

**Figure 1 pharmacy-08-00165-f001:**
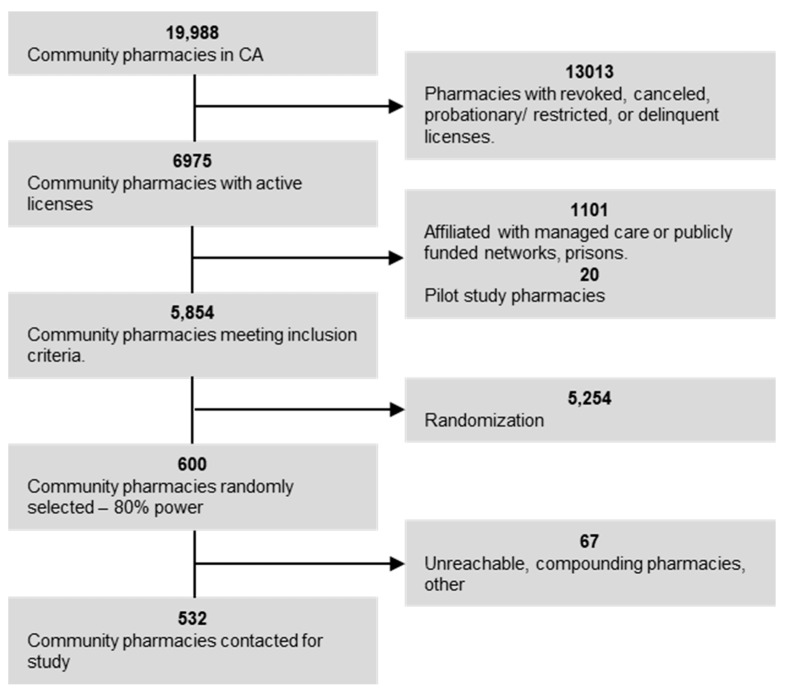
Pharmacy selection.

**Table 1 pharmacy-08-00165-t001:** Incomes by percentiles for Q2A–C.

	Q2A			Q2B			Q2C		
Income Percentile	No (%)	Yes (%)	Total (%)	No (%)	Yes (%)	Total (%)	No (%)	Yes (%)	Total (%)
**20th**	0 (0.0)	1 (0.6)	1 (0.6)	0 (0.0)	2 (1.2)	2 (1.2)	1 (0.6)	0 (0.0)	1 (0.6)
**40th**	16 (9.0)	40 (22.5)	56 (31.5)	27 (15.6)	24 (13.9)	51 (29.5)	31 (17.8)	19 (10.9)	50 (28.7)
**60th**	21 (11.8)	51 (28.6)	72 (40.5)	29 (16.8)	37 (21.4)	66 (38.2)	34 (19.5)	34 (19.5)	68 (39.1)
**80th**	9 (5.1)	34 (19.1)	43 (24.2)	20 (11.6)	31 (17.9)	51 (29.5)	26 (14.9)	22 (12.6)	48 (27.6)
**95th**	1 (0.6)	5 (2.8)	6 (3.4)	2 (1.2)	1 (0.6)	3 (1.7)	4 (2.3)	3 (1.7)	7 (4.0)
**Total**	47 (26.4)	131 (73.6)	178 (100)	78 (45.1)	95 (54.9)	173 (100)	96 (55.2)	78 (44.8)	174 (100)

**Table 2 pharmacy-08-00165-t002:** Survey questionnaire.

**Q1**	Have you heard of the new law allowing pharmacies to dispense a year-long supply of birth control at once?
**Q2A**	Would you be able to dispense a year-long supply of birth control at once to cash pay patients?
**Q2B**	Would you be able to dispense a year-long supply of birth control at once to privately insured patients?
**Q2C**	Would you be able to dispense a year-long supply of birth control at once to publicly insured patients?
**Q3**	What do you think are obstacles to dispensing a year-long supply of birth controls at once?

**Table 3 pharmacy-08-00165-t003:** Response to Question 3.

Obstacles to Dispensing Year-long Birth Control		
Obstacle	N (total = 532)	Percent (%)
Insurance	297	55.83%
Store Policy	71	13.35%
Supply	28	5.26%
Prescription Validity	16	3.01%
Unsure	15	2.82%
Legality	10	1.88%
Patient Safety	6	1.13%
Pharmacist comfort	3	0.56%
Medication Concerns	1	0.19%
N/A	25	4.70%

**Table 4 pharmacy-08-00165-t004:** Response to Question 1 and 2A, B, C.

Survey Results					
	Questions	Yes (%)	No (%)	N/A (%)	Total (*n*)
Q1	Have you heard of the new law allowing pharmacies to dispense a year-long supply of birth control at once?	177 (33.3)	353 (66.4)	2 (0.3)	532
Q2A	Would you be able to dispense a year-long supply of birth control at once to cash pay patients?	132 (72.9)	47 (26.0)	2 (1.1)	181
Q2B	Would you be able to dispense a year-long supply of birth control at once to privately insured patients?	95 (54.6)	78 (44.8)	1 (0.5)	174
Q2C	Would you be able to dispense a year-long supply of birth control at once to publicly insured patients?	78 (44.6)	97 (55.4)	0 (0)	175

**Table 5 pharmacy-08-00165-t005:** Yes or no responses to Question 1 compared to yes or no responses to Questions 2A–C.

Question 1 Compared to Questions 2A, 2B and 2C				
Questions		Q1 (Y)	Q1 (N)	*p*-value
Q2A	Yes	81%	70%	0.105
	No	19%	30%	
Q2B	Yes	48%	58%	0.213
	No	52%	42%	
Q2C	Yes	47%	43%	0.687
	No	53%	57%	

**Table 6 pharmacy-08-00165-t006:** Logistic regression model control for store policy, patient safety, and insurance.

Logistic Regression (LR)	Main Study Variable: Pay Type OR (95% CI)			Covariate(s)
	Cash vs Private	Cash vs Public	Private vs Public	Obstacle
LR model 1	2.82 (1.68, 4.72) ^#^	4.39 (2.63, 7.33) ^#^	1.56 (0.99, 2.46)	Store Policy: 0.03 (0.01, 0.07) ^#^
LR model 2	2.27 (1.44, 3.56) ^^^	3.50 (2.23, 5.48) ^#^	1.54 (1.01, 2.36)	Patient Safety: 0.09 (0.01, 0.50) ^&^
LR model 3	6.16 (3.45, 11.02) ^^^	10.07 (5.52, 18.35) ^#^	1.63 (1.04, 2.56) ^&^	Insurance: 5.30 (3.19, 8.83) ^#^
LR model 4	5.02 (2.62, 9.63) ^#^	8.29 (4.28, 16.06) ^#^	1.65 (1.03, 2.65) *	Store Policy: 0.04 (0.01, 0.10) ^#^
				Patient Safety: 0.06 (0.01, 0.30) ^&^
				Insurance: 2.59 (1.43, 4.82) ^&^

^#^*p* < 0.0001, ^^^
*p* < 0.001, ^&^
*p* < 0.01, * *p* < 0.05.

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
