# Peer review of "Pharmacy Implementation of a New Law Allowing Year-Long Hormonal Contraception Supplies"

_pharmacy, 2020, doi:10.3390/pharmacy8030165_

Round 1
Reviewer 1 Report
Thank you for the opportunity to review this manuscript. This study examined pharmacist knowledge of the new legislation of expanded access to contraception and their perceived behavior of dispensing yearlong contraceptives. The manuscript does a good job in demonstrating how pharmacies were selected and how secret-shopper surveys were conducted. The study had a some limitations that require consideration:
INTRODUCTION
What was encompassed in the comprehensive family planning services?
Please change Center to Centers for Disease Control and Prevention.
I think some additional information on the yearlong supply of chosen prescription contraceptive method would be helpful. For example, which contraceptive methods are covered – is it all?; Also, how was the law disseminated? Is it enforced or perceived to be optional?
METHODS
Secret shopper methodology
Agree with role theory as a framework but would like to see this explained a bit more (I anticipate the broader readership isn’t going to be familiar with role theory).
What was the primary endpoint used for the power calculation?
In Figure 1. What does the Randomization for the 5,254 mean?
It seems that you used a macro developed for SAS? Should cite this and state which tests this uses. This should also be with the rest of the statistical analysis section.
What was the approach for coding the qualitative variables? Was coding standardized across staff? Was there any cross-checking? And were staff trained?
The baseline/demographic characteristics need to be described earlier – prior to the analysis section - including how they were obtained.
It appears that obstacles were individually controlled in the multivariate analysis of payment type’s effect on contraceptive dispense. However, it’s very possible that listed obstacles are related to both pharmacist’s knowledge of new policy and their dispense behavior. Therefore, it will be important to know what the association would be like if all obstacles are controlled in the regression. Further, the question doesn’t ask whether there was an obstacle for them but rather what obstacles they perceive are possible. Either way, this regression requires much more detail in order to understand the approach. It would be important to understand how the qualitative variables were coded and included into the model and why some variables were omitted.
Question 3 was asked after Q2 A/B/C, meaning the obstacle information was acquired after the interviewers brought up insurance type. This might create bias for the results.
RESULTS
There are no descriptive statistics of the responding pharmacies.
Comparisons between type of insurance proportions are not shown in the results.
I don’t think the asterisks on Table 4 are needed as you’re already pointing to the question.
The regression results are hard for me to interpret as I do not understand the analytical approach (see comment above). A table would be helpful.
DISCUSSION
So, you calculated that you need 581 pharmacies, wanted to recruit 600, but ultimately ended up with fewer pharmacies. Why not continue going down the line of randomly ordered pharmacies until you reach recruitment goal?
Was it possible to collect characteristics about the pharmacy (location, independent vs chain, etc.)?
Author Response
INTRODUCTION
What was encompassed in the comprehensive family planning services?
Edit: added “ranging from providing birth control methods (long-acting reversible methods, birth control pills/patch/ring, sterilization), abortion care, testing and treatment for sexually transmitted disease, and cervical cancer screening,”
Please change Center to Centers for Disease Control and Prevention.
Edit: Centers for Disease Control and Prevention
I think some additional information on the yearlong supply of chosen prescription contraceptive method would be helpful. For example, which contraceptive methods are covered – is it all?; Also, how was the law disseminated? Is it enforced or perceived to be optional?
Edit: “All forms of self-administered prescription birth control methods – including the combined and progestin-only oral contraceptive pills, the vaginal ring, and the hormonal patch – were included in the law providing yearlong supply.”
METHODS
Secret shopper methodology
Edit: “Secret shopper” is an observational research methodology used to evaluate internal consistency among entities to identify areas for process improvement by using an individual surveyor with acting as a care provider asking about access to yearlong self-administered prescription birth control for her patients.”
Agree with role theory as a framework but would like to see this explained a bit more (I anticipate the broader readership isn’t going to be familiar with role theory).
Edit: Role theory is a sociological concept that hypothesizes that all interactions are played out in a circumscribed set of categories where each role has widely recognized rules, norms and expectations that each participant must act out and interact with (eg: moviegoer and usher taking tickets). Role theory is critical in qualitative research, especially with secret/mystery shopper methodology as the whole basis of the investigation rests on the subject performing their predefined role and the investigator challenging that role while remaining within bounds of what is considered "socially acceptable" for particular interaction (eg: moviegoer and usher telling you the weather forecast) [9]
What was the primary endpoint used for the power calculation?
I am not sure what you mean as “endpoint”. We determined that we should question 20 pharmacies using the same script template. From there, based on the responses, we found to have a power of 80%, the study needed to include 581 pharmacies. I have not made any edits to the text in regard to this comment.
In Figure 1. What does the Randomization for the 5,254 mean?
Explanation: there were 5854 pharmacies after exclusion of pharmacies. From those 5854, 600 were randomly selected to be called, leaving out 5254 pharmacies. So, 5254 were excluded due to randomization.
Edit: 600 were randomly selected from 5254 and those 600 were randomized to 3 groups.
It seems that you used a macro developed for SAS? Should cite this and state which tests this uses. This should also be with the rest of the statistical analysis section.
Explanation: in most results, we use logistic regression procedures in SAS. It wasn't SAS macro.
What was the approach for coding the qualitative variables? Was coding standardized across staff? Was there any cross-checking? And were staff trained?
Edit: This response was analyzed for content and keywords were tagged to allow thematic analysis of the qualitative data. These tagged keywords were then reviewed by the one data reviewer for standardization.
Explanation: The results were reviewed by each trained staff surveyor and keywords were tagged. These were all reviewed by one data reviewer who coded the responses in each of the 12 qualitative variable categories.
The baseline/demographic characteristics need to be described earlier – prior to the analysis section - including how they were obtained.
Edit: Pharmacies were stratified by income percentiles using their zipcodes. Income percentiles were derived from statiticalatlas.com using the data provided by US Census bureau and then divided into 20th, 40th, 60th, 80th and 95th percentiles. (Tables 1A-C).
Tables added (Table 1A-C)
It appears that obstacles were individually controlled in the multivariate analysis of payment type’s effect on contraceptive dispense. However, it’s very possible that listed obstacles are related to both pharmacist’s knowledge of new policy and their dispense behavior. Therefore, it will be important to know what the association would be like if all obstacles are controlled in the regression. Further, the question doesn’t ask whether there was an obstacle for them but rather what obstacles they perceive are possible. Either way, this regression requires much more detail in order to understand the approach. It would be important to understand how the qualitative variables were coded and included into the model and why some variables were omitted.
Edit: When we controlled for the other qualitative variables there was no notable statistical significance between whether the pharmacists would or would not dispense to different insurance types.
Explanation to how qualitative variables were coded is previously explained.
Question 3 was asked after Q2 A/B/C, meaning the obstacle information was acquired after the interviewers brought up insurance type. This might create bias for the results.
Explanation: To be clear, each pharmacist was asked one of three questions for Question 2. So, for example, they were either asked “Would you dispense yearlong birth control to cash-pay patients?” OR “Would you dispense yearlong birth control to privately insured patients” OR “would you dispense yearlong birth control to publicly insured patients?” They were then asked what their perceived obstacle would be. Thus, they did not bring up insurance type for Question 2, but then were allowed to bring up any obstacle they would perceive for question 3.
RESULTS
There are no descriptive statistics of the responding pharmacies.
Added, see above.
Comparisons between type of insurance proportions are not shown in the results.
Type of insurance proportions were divided at the start of the trial. Approximately 1/3 (181) pharmacists were asked if they would dispense to cash pay, 1/3 (174) pharmacists were asked if they would dispense to privately insured patients, 1/3 (175) pharmacists were asked if they would dispense to publicly insured patients.
I don’t think the asterisks on Table 4 are needed as you’re already pointing to the question.\
Asterisks removed.
The regression results are hard for me to interpret as I do not understand the analytical approach (see comment above). A table would be helpful.
Explained above.
DISCUSSION
So, you calculated that you need 581 pharmacies, wanted to recruit 600, but ultimately ended up with fewer pharmacies. Why not continue going down the line of randomly ordered pharmacies until you reach recruitment goal?
We were limited by time. Our goal was to complete questioning by 11/2017.
Was it possible to collect characteristics about the pharmacy (location, independent vs chain, etc.)?
The pharmacies names were listed as well as location. It would require some further review and data collection to be able to provide. Possibly can be included in subsequent follow-ups.

Reviewer 2 Report
This is an extremely important and very timely topic to study. Overall this is a very well-designed study. Minor suggestions below.
Line 38/39: I had to reread this sentence a few times to understand; consider adding "who live below the poverty level" or "living below" to make it read a little more easily.
Line 39: "The Affordable Care Act has attempted" personifies the act in a way that reads oddly to me - perhaps say that "The Affordable Care Act was passed in an attempt to.."
Line 56: Which population is African American women being compared to (all others? Caucasian?)
Line 115/116: Is there an additional phrase or missing word here? (amended to read goes on to clarify)
Line 118-120: Consider: "With the passage of CSB 999, pharmacists can dispense twelve pill packs per patient request even if..."
Line 273: nitpicky, but would "prescriber" be better used here - I know CA considers pharmacists providers so the use of this term could technically include them3
Author Response
This is an extremely important and very timely topic to study. Overall this is a very well-designed study. Minor suggestions below.
Line 38/39: I had to reread this sentence a few times to understand; consider adding "who live below the poverty level" or "living below" to make it read a little more easily.
Edit made to "Notably the rate of unintended pregnancies in women who live below the United States national poverty line is five times the rate in women 200% above the poverty line [1]."
Line 39: "The Affordable Care Act has attempted" personifies the act in a way that reads oddly to me - perhaps say that "The Affordable Care Act was passed in an attempt to.."
Edit made to: "The Affordable Care Act was passed in an attempt to remove obstacles to access by decreasing or eliminating patient co-pays for prescription contraception services and supplies for many insured patients. "
Line 56: Which population is African American women being compared to (all others? Caucasian?)
Edit made to: "In a study performed by Barber et al in 2019, while African American women tend to live closer to pharmacies than their white counterparts, those pharmacies are more likely to be community based, independent pharmacies that have limited opening hours, less visible contraceptive information, more difficult access to condoms (behind the counter) and fewer self-checkout options [2]."
Line 115/116: Is there an additional phrase or missing word here? (amended to read goes on to clarify)
Edit made to: "CSB 999 is a bill that amends and adds to previous laws: amendment to Section 4064.5 of the Business and Professions Code; amendment toSection 1367.25 of the Health and Safety Code; amendment to Section 10123.196 of the Insurance Code; addition to Section 14000.01 to the Welfare and Institutions Code."
Line 118-120: Consider: "With the passage of CSB 999, pharmacists can dispense twelve pill packs per patient request even if..."
Edit made to: "With the passage of CSB 999, pharmacists can dispense twelve pill packs per patient request even if the prescription is written for a 3-month supply with refills."
Line 273: nitpicky, but would "prescriber" be better used here - I know CA considers pharmacists providers so the use of this term could technically include them
Edit made to: prescribers from providers throughout the manuscript.
Thank you for your corrections and recommnedations.
Round 2
Reviewer 1 Report
Thank you for your revisions. I feel that these have strengthened the manuscript considerably. Please see below for some follow-up comments.
The explanation on Role Theory is great - thank you. I think you could remove the moviegoer example.
Regarding the sample size calculation - I am interested in what question or endpoint was used from those initial pilot pharmacies to calculate the effect size for the sample size estimation. Clarification is needed here.
I get the sense that the qualitative approach was inductive - is this correct?
There is still very little information on the logistic regression approach and I'm not able to follow it. Were these a series of simple models or was there a multiple regression model? I also reiterate my suggestion for a table of these results.
Tables 1 and 3 should go in the results, if possible.
Author Response
Thank you again for your thoughtful critique. Here are the edits made in regards to your recommendations:
The explanation on Role Theory is great - thank you. I think you could remove the moviegoer example.
Edit: “Role theory is a sociological concept that hypothesizes that all interactions are played out in a circumscribed set of categories where each role has widely recognized rules, norms and expectations with which each participant must act out and interact.”
Regarding the sample size calculation - I am interested in what question or endpoint was used from those initial pilot pharmacies to calculate the effect size for the sample size estimation. Clarification is needed here.
Edit: “Among randomly selected 20 pharmacists, 100% , 42.9% and 28.6% answered they would dispense a yearlong supply to cash-pay, privately and publicly insured patients respectively. Using these proportions as preliminary data, we powered (80% power) for all pairwise comparisons at 0.017 significance level.”
I get the sense that the qualitative approach was inductive - is this correct?
Response: correct, we used an inductive approach to determine keywords and obstacles from responses provided by pharmacists.
There is still very little information on the logistic regression approach and I'm not able to follow it. Were these a series of simple models or was there a multiple regression model? I also reiterate my suggestion for a table of these results.
Edit: “When comparing the data presented in Question 3 against the three cohorts from Question 2, there were several areas of correlation with statistical significance. A pharmacist was 6.16 times more likely to dispense a one-year supply of birth control to cash-pay compared to privately insured patients (CI 95%, 3.45 – 11.02, p < .001) and 10.07 times more likely to cash-pay compared to publicly insured patients (CI 95%, 5.52 – 18.35, p < 0.0001) when we controlled for the variable that stated whether or not insurance was an obstacle. Furthermore, they were 1.63 times more likely to dispense a one-year supply of birth control to privately insured patients compared to publicly insured patients (CI 95%, 1.04 – 2.56, p <0.01) when we controlled for the variable that stated whether or not insurance was an obstacle. When we controlled for the variable that stated whether or not store policy was an obstacle, pharmacists were 2.82 times more likely to dispense a one-year supply of birth control to cash pay compared to privately insured patients (CI 95%, 1.68 – 4.72, p <0.0001) and 4.39 times more likely to cash-pay compared to publicly insured patients (CI 95%, 2.63 – 7.33, p < 0.0001). Similarly, when we controlled for the variable that stated whether or not patient safety was an obstacle, pharmacists were 2.27 times more likely to dispense a one-year supply of birth control to cash pay compared to privately insured patients (CI 95%, 1.44 – 3.56, p < 0.001) and 3.50 times more likely to cash-pay compared to publicly insured patients (CI 95%, 2.23 – 5.48, p < 0.0001). Lastly, we also incorporated store policy, patient safety and insurance in the model along with the main study variable 'pay type' we found that a pharmacist was 5.02 times more likely to dispense a one-year supply of birth control to cash-pay compared to privately insured patients (CI 95%, 2.62 – 9.63, p < .0001) and 8.29 times more likely to cash-pay compared to publicly insured patients (CI 95%, 4.28 – 16.06, p < 0.0001) when we controlled for all three variables (whether or not store policy, patient safety and insurance were obstacles). Furthermore, they were 1.65 times more likely to dispense a one-year supply of birth control to privately insured patients compared to publicly insured patients (CI 95%, 1.03 – 2.65, p < 0.05). (Table 6: Logistic Regression Model)”
Additional table
|
Table 6: Logistic regression model |
||||
|
Logistic regression (LR) |
Main study Variable: Pay Type OR (95% CL) |
Covariate(s) |
||
|
Cash vs Private |
Cash vs Public |
Private vs Public |
Obstacle |
|
|
LR model 1 |
2.82 (1.68, 4.72)# |
4.39 (2.63, 7.33)# |
1.56 (0.99, 2.46) |
Store Policy: 0.03 (0.01, 0.07)# |
|
LR model 2 |
2.27 (1.44, 3.56)^ |
3.50 (2.23, 5.48)# |
1.54 (1.01, 2.36) |
Patient Safety: 0.09 (0.01, 0.50)& |
|
LR model 3 |
6.16 (3.45, 11.02)^ |
10.07 (5.52, 18.35)# |
1.63 (1.04, 2.56)& |
Insurance: 5.30 (3.19, 8.83)# |
|
LR model 4 |
5.02 (2.62, 9.63)# |
8.29 (4.28, 16.06)# |
1.65 (1.03, 2.65)* |
Store Policy: 0.04 (0.01, 0.10)# |
|
Patient Safety: 0.06 (0.01, 0.30)& |
||||
|
Insurance: 2.59 (1.43, 4.82)& |
||||
Tables 1 and 3 should go in the results, if possible
Edit: correction made, moved to results, see text.
Please let me know if there are any further corrections or edits to be made.
Round 3
Reviewer 1 Report
Thank you for the continued adjustments to the paper. One more set of comments. Regarding the logistic regression model, I'm interpreting it as follows:
For the LR Model 1, there was one model constructed comparing the three types of insurance types on the willingness of pharmacists to dispense a 1-year supply, controlling for store policy. Similar logic applies to #2 and #3. #4 is similar but controls for store policy, patient safety, and insurance.
If this is accurate, then I think we're good. The only thing that I would ask is a descriptive title for the Logistic Regression Table.
Author Response
For the LR Model 1, there was one model constructed comparing the three types of insurance types on the willingness of pharmacists to dispense a 1-year supply, controlling for store policy. Similar logic applies to #2 and #3. #4 is similar but controls for store policy, patient safety, and insurance.
If this is accurate, then I think we're good. The only thing that I would ask is a descriptive title for the Logistic Regression Table.
Comment: You are correct. # 1, 2 and 3 control for insurance, patient safety and store policy respectively. In # 4 it controls for all three.
I have added this descriptive title: Logistic Regression Model Control for Store Policy, Patient Safety, and Insurance
Thank you again for your helpful comments.